# The ATM Ser49Cys Variant Effects ATM Function as a Regulator of Oncogene-Induced Senescence

**DOI:** 10.3390/ijms25031664

**Published:** 2024-01-29

**Authors:** Caroline Atkinson, Aideen M. McInerney-Leo, Martina Proctor, Catherine Lanagan, Alexander J. Stevenson, Farhad Dehkhoda, Mary Caole, Ellie Maas, Stephen Ainger, Antonia L. Pritchard, Peter A. Johansson, Paul Leo, Nicholas K. Hayward, Richard A. Sturm, Emma L. Duncan, Brian Gabrielli

**Affiliations:** 1Mater Research Institute, The University of Queensland, Brisbane, QLD 4102, Australia; 2Dermatology Research Centre, Frazer Institute, The University of Queensland, Brisbane, QLD 4102, Australia; 3Queensland Institute for Medical Research Berghofer, Brisbane, QLD 4006, Australia; 4Centre of Genomics and Personalised Health, Queensland University of Technology, Brisbane, QLD 4059, Australia; 5Department of Twin Research and Genetic Epidemiology, School of Life Course & Population Sciences, Faculty of Life Sciences and Medicine, King’s College London, London SE1 1UL, UK

**Keywords:** ATM, p53, ionising radiation, DNA damage, senescence

## Abstract

An apical component of the cell cycle checkpoint and DNA damage repair response is the ataxia-telangiectasia mutated (ATM) Ser/Thr protein kinase. A variant of ATM, Ser49Cys (rs1800054; minor allele frequency = 0.011), has been associated with an elevated risk of melanoma development; however, the functional consequence of this variant is not defined. ATM-dependent signalling in response to DNA damage has been assessed in a panel of patient-derived lymphoblastoid lines and primary human melanocytic cell strains heterozygous for the ATM Ser49Cys variant allele. The ATM Ser49Cys allele appears functional for acute p53-dependent signalling in response to DNA damage. Expression of the variant allele did reduce the efficacy of oncogene expression in inducing senescence. These findings demonstrate that the *ATM 146C>G* Ser49Cys allele has little discernible effect on the acute response to DNA damage but has reduced function observed in the chronic response to oncogene over-expression. Analysis of melanoma, naevus and skin colour genomics and GWAS analyses have demonstrated no association of this variant with any of these outcomes. The modest loss of function detected suggest that the variant may act as a modifier of other variants of ATM/p53-dependent signalling.

## 1. Introduction

DNA damage can be induced by a variety of sources such as ionising radiation and ultraviolet light. Cell cycle checkpoints are responsible for co-ordinating cell cycle and repair responses to such DNA damage. Defects in these checkpoints can result in the accumulation of DNA damage, leading to genomic instability and increased susceptibility to cancer development. Ataxia-telangiectasia mutated (ATM) is an apical component of the DNA damage signalling pathways involved in cell cycle checkpoint and DNA repair. In response to DNA double-strand breaks (DSBs) caused by ionising radiation, ATM is activated by autophosphorylation at S1981 [1,2]. Several substrates are phosphorylated by ATM such as p53, CHK2 and BRCA1 which then trigger cell cycle arrest at G1, S and G2 phases and/or promote DNA repair [3]. In addition to immediate DNA damage responses, ATM has also been demonstrated as having a role in regulating the oncogene-induced senescence barrier to transformation [4]. Oncogene-induced senescence is a significant barrier to transformation in melanocytes, particularly as NRAS and BRAF oncogenic mutations are common driver of naevus formation and are also common oncogenic drivers of melanoma [5,6]. Thus, defective ATM function could contribute to bypassing the senescence barrier in oncogene expressing nevocytes.

Complete ATM loss of function caused by biallelic *ATM* pathogenic variants results in the disease ataxia telangiectasia (AT), characterized by immunodeficiency, neurological degeneration and predisposition to cancer [7]. Epidemiological studies show that obligate heterozygote parents of individuals with AT reveal an increased risk of cancer, which may be true of *ATM* heterozygotes in the general population [8,9,10,11].

There are over 500 reported population variants of *ATM* (gnomAD; https://gnomad.broadinstitute.org/ (accessed on 21 November 2021)); however, most are of unknown or uncertain effect. A large multicentre study of familial, multiple primary (MPM) and sporadic melanoma cases found a significant association with *ATM* germline variants of unknown significance (VUS) suggesting these conferred a moderately increased risk of developing melanoma. One of these is the *ATM* missense variant c.146C>G (Ser49Cys, rs1800054 Ser TCC to Cys TGC) that is present in ~2.2% of the population (gnomAD). This variant has been reported to confer an increased risk of melanoma in a large Danish population [12]. It is, however, defined as ‘benign’ in ClinVar (VCV000003048) and although *ATM* SNPs are associated with both cancer types in GWAS studies, this variant has not reached significance [11,13]. This variant is located in a region of ATM that is required for binding of p53, a phosphorylation target of ATM in its response to DNA damage [14]. We have previously identified a family with a history of early-onset melanoma and astrocytoma with an allelic loss of function of CDKN2A p14ARF-specific transcript mutation (c.193G>A; p.G65S) and the ATM Ser49Cys allele [15]. The p14ARF mutation resulted in loss of expression of the mutant allele which has been associated with increased susceptibility to melanoma and neural cancers [16]. The carriers of both the p14ARF and ATM variant alleles had aggressive, early-onset disease suggesting that the ATM variant may be a modifier of p14ARF predisposition in those individuals [15]. However, at present, there is no evidence that Ser49Cys substitution affects ATM protein activity or function. This study investigates the functional consequence of ATM Ser49Cys substitution in ATM-dependent responses to acute DNA damage and chronic oncogene-induced stress that promotes senescence. As the variant allele is normally present in a heterozygous state with wild-type ATM, the effect of variant expression in the ATM wild-type background was assessed to determine whether any effects of the variant could be detected in this heterozygous wild-type-variant background.

## 2. Results

### 2.1. ATM Ser49Cys Variant Protein Is Functional for Acute p53 Activation by DNA Damage

The ATM Ser49Cys variant is relatively common as a heterozygote in the human population (2.2%). To assess the functional outcome of the variant, we attempted to use CRISPR to knockin the variant in HT1080 human fibrosarcoma cells. Despite attempts with multiple gRNAs and approaches and sequencing multiple clones, no knockins of the SNP were found. This was not because the homozygous variant was non-viable as 132 individuals in 815,000 exomes are homozygous for this variant; the frequency of homozygotes in the population is consistent with the minor allele frequency (0.00011 het vs. 0.000162 homo; https://gnomad.broadinstitute.org/variant/11-108227849-C-G?dataset=gnomad_r4 (accessed on 21 November 2021)). While CRISPR creation of a homozygous ATM Ser49Cys variant was not possible in our hands and its infrequency as a homozygote in the population (0.02%) meant we were unable to identify any in our cell line bank, we were able to make use of naturally occurring heterozygote cell lines for our functional investigation. We utilised a panel of six lymphoblastoid cell lines (LCL), two wild types for ATM including an unaffected relative of the proband from the affected family (Col1.201, carrier of the CDKN2A variant but wild type for ATM), an ATM null mutant (AT3ABR) and three ATM Ser49Cys heterozygous allele carriers including one from another unaffected relative of the proband (Col 1.202, wild type for CDKN2A and heterozygous for Ser49Cys ATM). We also used six melanoblast cell strains derived from ATM wild types (three lines, WT) and three Ser49Cys heterozygotes. The genotypes for the lines used are provided in Appendix A.

An initial time course analysis after ionising radiation (IR) exposure revealed little difference in the timing of either ATM phosphorylation on its activating S1981 site or CHK2 on T68 in the ATM wild-type JA line or the ATM variant 1.202 line. Both phosphorylations were detected and near maximal with 1–2 h post-irradiation (Figure 1A). Quantitative analysis of components of the ATM signalling pathway was performed at 1 h post-IR. Although there was cell line-to-cell line variation, the Ser49Cys variant protein was expressed at similar levels to the wild-type ATM in LCLs (Figure 1B–D), and the level of ATM activation indicated by the increased autophosphorylation at S1981 was also similar between the WT and Ser49Cys variant.

Multiple downstream ATM targets were analysed to assess ATM signalling. The levels of total p53 and phospho-p53 S15 (pp53), a direct ATM phosphorylation target, increased following irradiation in both ATM wild-type and Ser49Cys variants but not in the ATM null variant, although the level of increase varied in each line (Figure 1B–D). Even in the 1.202 LCL with the lowest level of p53 activation, the effect on p53 phosphorylation and accumulation was lost at longer time points (Appendix A). Phosphorylation of another ATM target CHK2 T68 was also increased following irradiation in all cell lines. The level of allelic expression was similar amongst the different LCL lines, with the *c.146G* allele representing >40% of the ATM expressed (Figure 1E).

A similar lack of effect of the ATM Ser49Cys allele on responses to IR was observed on a panel of human neonatal melanoblasts. A time course showed that ATM was activated with similar kinetics of ATM or CHK2 phosphorylation/activation to those of the LCLs (Figure 2A). The level of ATM protein and ATM activation with IR varied between melanoblast strains, but there were no significant differences between the wild-type and variant-expressing strains (Figure 2B,C). P53 phosphorylation appeared marginally reduced in IR-treated variant-expressing melanoblasts, but CHK2 activation was little affected, similar to what was observed with the LCLs.

The 1.202 LCL lines showed the lowest p53 activation in response to IR of the Ser49Cys variant-expressing lines and this was also seen in its downstream signalling, with more modest increases in mRNA expression of a panel of p53-regulated genes in response to IR including *CDKN1A*, *BBC3* and *GADD45A* (Figure 3A) and at the protein level for p21 (Figure 1B,C). However, this appeared to represent the variation observed across the entire panel of wild-type and Ser49Cys variants, and in all cases, the p53-regulated genes were regulated in a similar manner to the wild type. Using Nutlin 3a to inhibit MDM2 and stabilise p53, the level of p53 increase and of its transcriptional target p21 also varied with each cell line (Figure 3B), indicating that cell line-specific effects were contributing to the difference in p53 responses with IR treatments.

The effect of the variant on the DNA damage repair responses was also assessed by examining DNA damage foci formation after irradiation. In both LCL and melanoblast lines, BRCA1, 53BP1 and γH2AX focus formation was unaffected by Ser49Cys variant expression (Figure 4A; Appendix A). Using a p-ATM/p-ATR substrate antibody revealed little difference in the level or pattern of either immunoblotting or immunofluorescent staining of the variants compared to wild-type ATM-expressing cells; the ATM null cells were substantially different (Figure 4B; Appendix A). These data indicate that the Ser49Cys substitution has little effect on the phosphorylation of ATM substrate proteins.

### 2.2. ATM Ser49Cys Variant Delays Senescence in Primary Human Melanoblasts

ATM has a major role in the establishment of oncogene-induced senescence, in part through its activation of p53 [4]; and oncogenic *BRAF* and *NRAS* expression in skin melanocytes are primary drivers of senescence associated with naevus formation [5]. To investigate whether the ATM Ser49Cys variant was less susceptible to oncogene-induced senescence, six early-passage (passages 5–8) melanoblast lines (three ATM WT and three Ser49Cys) were transduced with an empty vector or mutant NRAS Q61K. After two weeks, cells were stained for either SAβ-Gal or immunostained for Ki67 as a marker of proliferation or γH2AX as a DNA damage marker. SAβ-Gal staining of the vector control-transduced melanoblasts showed that the WT but not the Ser49Cys variants were strongly senescent by this marker (Appendix A). The Ser49Cys melanoblasts were more resistant to senescence with >40% not staining for SAβ-Gal compared to 20% for the empty vector-transduced cells (Figure 5A; Appendix A). There was a concomitant decrease in the proportion of Ki67-positive cells in the NRAS-transduced WT melanocytes below the level of the empty vector-transduced cells (Figure 5B). By contrast, in all three Ser49Cys lines, NRAS transduction increased Ki67-positive staining, indicating that it was driving increased proliferation in these lines. Interestingly, the empty vector-transduced Ser49Cys melanoblasts all had relatively low levels of Ki67 staining despite having little SAβ-Gal staining. Both the vector and NRAS Q61K were expressed in >90% of the cells (Appendix A). This suggests that while the vector-transduced cells were not senescent, they were not strongly proliferative. NRAS transduction produced similar levels of DNA damage as measured by γH2AX focus formation in both WT and Ser49Cys melanoblast lines, indicating that it was the differing response to this damage that determined the level of senescence observed (Figure 5C). When the NRAS-transduced cultures were allowed to continue in culture, there was little change in the cell density of the WT lines, although these cells were viable, indicated by strong tetramethylrhodamine ethyl ester (TMRE) staining, demonstrating mitochondrial viability. In the Ser49Cys cultures, small colonies appeared that were strongly GFP- and TMRE-positive, indicating the outgrowth of proliferative cells (Figure 5D; Appendix A).

### 2.3. ATM Ser49Cys Is Not Associated with Increased Melanoma Risk

The limited data about the increased risk of melanoma associated with the ATM Ser49Cys variant are from a single study [12]. The very modest functional effects of the variant in its normal heterozygous setting identified here did not support the increased risk reported in the previous study. We investigated two separate datasets, the Australian Melanoma Genome Project (AMGP) and the Brisbane Naevus Morphology Study (BNMS [17]). The AMPG identified the Ser49Cys variant in the germline of 10 of 575 patients, i.e., 10 of 1150 alleles carry the variant (VAF = 0.87%), which is comparable with the VAF of 1. 1% in normal population (one-sided binomial test *p* = 0.30). In one of the AMGP melanomas from these germline carriers, ATM was hemizygous with the variant allele lost. In the BNMS dataset, 1255 volunteers were genotyped, with 22 individuals that carry the Ser49Cys variant as heterozygotes identified: VAF of 0.88%, 11 with melanoma, and 11 were unaffected. Whole-exome sequencing (WES) was performed on 384 high-risk unrelated individuals previously diagnosed with primary melanoma: n = 230 multiple primary melanoma (MPM) and n = 154 single primary melanoma (SPM) [18]. Chi-square analysis confirmed that the frequency of ATM Ser49Cys was significantly higher in a high-risk melanoma cohort (9/384 carriers; n = 7 multiple primary melanoma, n = 2 single primary melanoma) compared to the European cohort (non-Finnish) in GnomAD (*p* = 0.01). However, no statistical significance was observed between the frequency of this variant in melanoma cases vs. controls in the UK Biobank (https://biobankengine.stanford.edu/ (accessed 21 November 2021)). Together, these data do not support any predisposition of carriers of the ATM Ser49Cys allele for melanoma. Naevus number and size were also assessed in the BNMS cohort but this did not reveal any associations with Ser49Cys allele carriers. We also analysed the GeneATLAS [19] associations database for associations of the SNP with melanoma, other skin neoplasms and skin colour and found no significant associations (Table 1). Together, these data do not support any predisposition of carriers of the ATM Ser49Cys allele for melanoma or to alteration in an individual’s naevus properties.

## 3. Discussion

DNA damage response and repair pathways are critical in maintaining the genomic integrity of cells. ATM is a key protein involved in the DNA damage response (DDR) leading to cell cycle arrest or apoptosis [20]. ATM has been established as a predisposition gene for multiple malignancies including breast and pancreatic cancer and melanoma [8]. The ATM Ser49Cys variant was suggested from our previous familial melanoma and astrocytoma study as being a modifier of impact of a *CDKN2A* p14ARF variant, associating with an earlier age of onset [15]. The p14ARF variant reduced expression of p14ARF [16] which operates in response to stress by inhibiting MDM2 function and thereby p53 stability and function [21].

ATM variants located in the kinase domain may have adverse effect on kinase activity and/or impact on cell survival compared to variants located upstream of the kinase domain [22]. In a study assessing the function of multiple ATM variants of unknown significance, ATM Ser49Cys exhibited reduction in homology-directed repair with non-significant defects in ATM phosphorylation targets in ovarian cancer [8]. There was little obvious effect on p53 phosphorylation in response to IR suggesting that if the Ser49Cys mutation is influencing p53 binding, the effect is modest and, in the heterozygous wild-type-variant background, appears functional for acute DDR signalling including p53 phosphorylation. It is, however, impaired in its role in response to chronic stress imposed by oncogene expression that regulates progression into senescence. There was cell line-to-cell line variation in the acute DDR response in which the range of responses overlapped significantly. However, we identified reduced efficiency of the Ser49Cys variant in promoting senescence, indicating that signalling in response to chronic stress was reduced. Intriguingly, while oncogene-induced senescence of cutaneous melanocytes is responsible for naevus formation [5], this SNP (or others in *ATM*) has not been reported in any of the GWAS studies for melanoma or naevus susceptibility [23,24], and our analysis did not identify any significant association with naevus features in 22 carriers in the BNMS datasets.

Of interest, the MAF of ATM Ser49Cys (MAF = 0.007), is comparable to that of the known moderate-penetrance melanoma MITF germline variant Glu318Lys (MAF = 0.001) and also the suspected GOLM1 modifier variant Ser307Leu (MAF = 0.003). ATM Ser49Cys, GOLM1 Ser307Leu and MITF Glu318Lys all occur in higher frequencies in melanoma-affected populations (2.3%, 2.9% and 1.6–2.8%, respectively) as compared to general populations (~0.2% and 0.6–0.8%, respectively) [25,26]. However, while analysis of the UK Biobank data reveals that although the ATM p.S49C is not more frequent in melanoma cases as compared to controls, the MITF Glu318Lys variant is significantly more frequent is cases versus controls (*p* = 1.15 × 10^−10^). This may suggest that comparable to GOLM1 Ser307Leu, the ATM Ser49Cys variant is associated with a lower risk of melanoma and/or a later age of onset. Alternatively, genetic background may modify penetrance of the ATM Ser49Cys variant.

## 4. Materials and Methods

### 4.1. Antibodies

Mouse p53 clone DO-1 (used at 1/200 dilution for immunoblotting) was purchased from Santa Cruz Biotechnology, Dallas, TX, USA; antibodies against phospho-p53 S15 (used at 1/500 dilution for immunoblotting), CHK2 phospho-CHK2 T68 (used at 1/500 dilution for immunoblotting), pATM/ATR substrate (used at 1/200 dilution for immunoblotting and 1/100 for immunofluorescence) and phospho-ATM S1981 (used at 1/1000 dilution for immunoblotting) were from Cell Signalling Technology, Danvers, MA, USA. Antibodies against Cip1/WAF-1/p21 were from Upstate, Lancaster, NY, USA (used at 1/100 dilution for immunoblotting), ATM (used at 1/1000 dilution for immunoblotting) and rabbit RAD51 antibodies (used at 1/200 dilution for immunofluorescence) were from Calbiochem, San Diego, CA, USA, Ki-67 from Dako, Santa Clara, CA, USA, and CHK1 antibody were from Abcam, Cambridge, UK.

### 4.2. Production of Patient-Derived LCLs

Epstein–Barr virus (EBV)-immortalised lymphoblastoid cell lines (LCLs) were established from individuals by exogenous virus transformation of 2 × 10^6^ peripheral blood lymphocytes by using B95-8 EBV and were cultured in Dulbecco’s minimal essential medium with 10% calf serum supplemented with 20 mM Hepes pH 7.6. Peripheral blood lymphocytes were obtained from unaffected relatives of the proband from the melanoma and astrocytoma affected family with either wild-type or Ser49Cys ATM alleles [15].

### 4.3. Cell Culture

Patient-derived lymphoblastoid cell lines (LCL) JA, AT3ABR, Coll 1.201, Coll 1.202, A06 and D38 were maintained grown in Dulbecco’s minimal essential medium with 10% calf serum supplemented with 20 mM Hepes pH 7.6, were gifts from Professor Nickolas Hayward, QIMRB, Australia. LCL genotypes are provided in Appendix A. Primary human melanoblast cell strains QF1354, QF1363, QF1559, QF1433, QF1579 and QF1584 were maintained as previously described [27] and were gifts from Professor Richard Sturm, UQ, Australia. Polymorphisms within *ATM* and *MC1R* were genotyped as described below (Appendix A). Melanoblast strains and LCLs were exposed to 6 Gy γ-irradiation using a Gammacell 40 Exactor (Best Theratronics, Kanata, ON, Canada). Cells were analysed by immunofluorescence or underwent whole cell lysis and immunoblot analysis. Cell pellets were lysed on ice using NETN–lysis buffer (20 mM Tris-HCl pH8, 2 mM EDTA, 100 mM NaCl, 1% NonidetP40) containing 25 mM NaF, 25 mM b-glycerophosphate, protease inhibitor mix and 0.1% sodium dodecyl sulphate (SDS). Lysates were centrifuged at 4 °C, equalized for protein content, then boiled in SDS sample buffer. Proteins were resolved by SDS-polyacrylamide gel electrophoresis then transferred PVDF membrane using a Bio-Rad TransBlot^®^ SD SemiDry Transfer Cell (Bio-Rad Laboratories Pty Ltd., Gladesville, NSW, Australia) for 20 min at 15 V constant voltage. Membranes were initially blocked using 5% skim milk powder in Tris-buffer saline and 0.1% Tween-20 (TBST) then probed with appropriate primary and secondary antibodies using TBSTB washes. Protein bands were visualized by chemiluminescence using Western Lightning Plus Enhanced Chemiluminescence (Perkin-Elmer, Waltham, MA, USA).

### 4.4. Genotyping of LCLs and Primary Human Melanoblasts

Genomic DNA prepared from these cells was genotyped using the Illumina Infinium Microarray HumanCoreExomev24 Chip (Illumina Inc., Foster City, CA, USA) as described [28] and with SNPs not available on the Chip genotyped using TaqMan SNP genotyping assays [29] in 96- or 384-well plate format using a 7500 real-time PCR system and analysed using 7500 Software (Applied Biosystems, Foster City, CA, USA).

### 4.5. Allelic Expression Variation

Genomic DNA was isolated from LCL JA using the ISOLATE II Genomic DNA Kit (Bioline, London, UK) following the manufacturer’s instructions. RNA was isolated from LCL JA, D38, A06 and Coll 1.202 using Trizol, following the manufacturer’s instructions, then converted to cDNA with Superscript IV (Invitrogen, Waltham, MA, USA). cDNA and gDNA samples were analysed by the Australian Genome Research Facility (Nedlands, WA, USA). cDNA nucleotide percentages graphed using GraphPad Prism version 7.0.

### 4.6. Immunofluorescence

1 × 10^5^ Melanoblasts were grown on Poly-L-lysine (Sigma Aldrich, St. Louis, MI, USA)-coated coverslips for 48 h then irradiated as above and, 2 h later, fixed with 4% paraformaldehyde. LCLs were cytospun onto glass slides (800 rpm for 5 min) at 2 h post-irradiation then fixed in 4% PFA. Coverslips were permeabilized using 0.1% Triton X100 for 15 min then blocked with 3% BSA. Primary antibodies were diluted in blocking solution and washes performed using TBS 0.1% skim milk powder. Coverslips were mounted onto glass slides using Prolong Gold mounting media (Invitrogen, Waltham, MA, USA) and imaged using an Olympus BX63 upright microscope (Olympus Corporation, Tokyo, Japan).

### 4.7. Induction of Senescence

Melanoblast cells were seeded at 3000 cells/well in 96-well plates (Corning) then transduced with either NRAS Q61K containing virus or vector control (pLV411, Genebank ID pLVEiG). Plates were incubated for 14 days with media changes every 4 days. One plate was fixed and stained for senescence-associated β-galactosidase (SA-β-Gal), and two plates were fixed and stained for quantitative immunostaining for Ki67 and γH2AX using Cell Profiler [30], and analysis was performed using R-Studio.

### 4.8. Statistical Analysis

Analysis was performed using an unpaired t-test in Prism GraphPad version 7.0.

## 5. Conclusions

Heterozygous expression of the common ATM Ser49Cys variant did not affect acute signalling in response to DNA damage but was less effective in promoting senescence in response to oncogene expression. These data indicate that as a single variant, ATM Ser49Cys is benign, in agreement with its classification in ClinVar, but the early onset of cancers in probands carrying the ATM Ser49Cys and CDKN2A p14ARF variants suggest that the ATM variant may act as a modifier of risk via impaired senescence. Even modest reduction in ATM function may be enough to impair its role in regulating p53 function sufficiently to permit enhanced genome instability in conjunction with the CDKN2A p14ARF variant, resulting in enhanced tumour progression and aggressiveness.

## Figures and Tables

**Figure 1 ijms-25-01664-f001:**
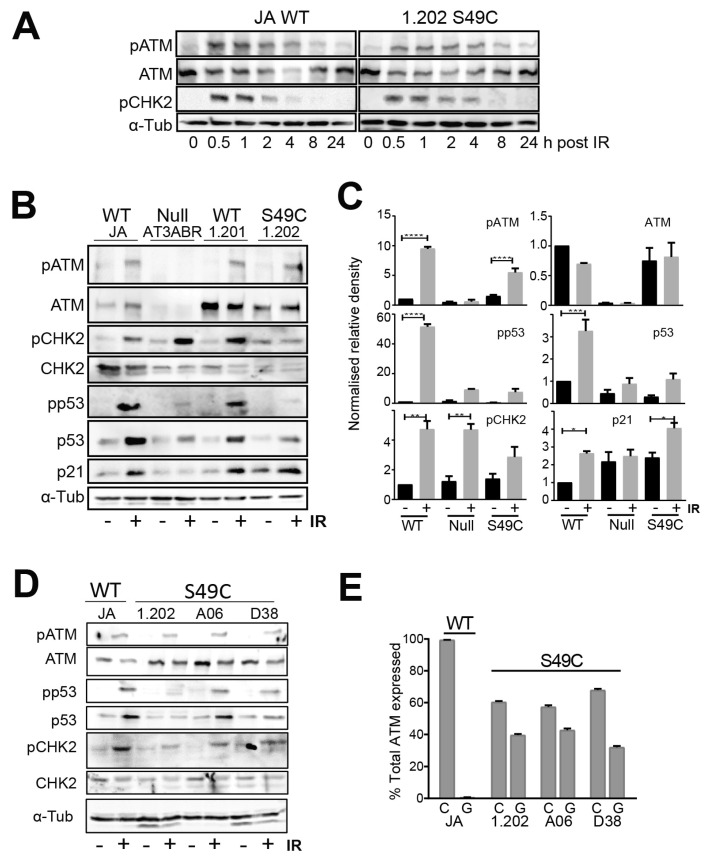
ATM Ser49Cys variant LCL lines have functional p53-dependent responses. (**A**) The indicated LCL lines were irradiated with 6Gy IR and then harvested at the indicated times. Whole cell lysates were immunoblotted for ATM, phospho-ATM S1981 (pATM), phospho-CHK2 T68 (pCHK2) and α-tubulin as a loading control. This is representative of three independent experiments. (**B**) The indicated LCL cell lines were cultured and treated with or without 6 Gy IR for 1 h. Whole cell lysates were immunoblotted for ATM, pATM, CHK2, pCHK2, p53, phospho-p53 S15 (pp53), p21 and α-tubulin as a loading control. Figure is a representation of four independent experiments. (**C**) Densitometric quantification of immunoblots, relative to untreated JA (WT), normalised by α-tubulin. Error bars indicate standard error of the mean of three biological replicates. * *p* < 0.05, ** *p* < 0.01, *** *p* < 0.001, **** *p* < 0.0001. (**D**) Whole cell lysates from LCL cell lines harvested at 1 h post-irradiation with 6 Gy IR were immunoblotted with indicated antibodies. α-tubulin was used as a loading control. This is representative of two independent experiments. (**E**) Allele-specific expression of ATM146 G, C alleles for the ATM WT JA and three ATM variant lines. The data are the mean and SD for three replicates and are representative of two independent experiments.

**Figure 2 ijms-25-01664-f002:**
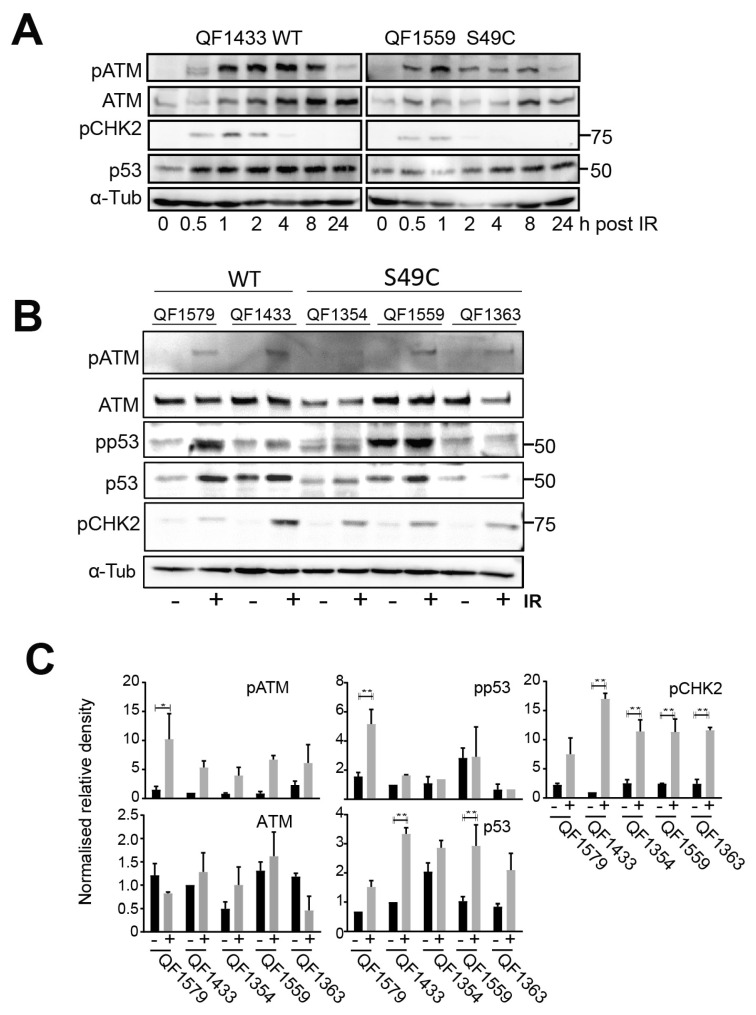
ATM Ser49Cys variant melanoblast lines have functional p53-dependent responses. (**A**) The indicated human melanoblasts were irradiated with 6 Gy IR, then harvested at the indicated times. Whole cell lysates were immunoblotted for the indicated proteins. (**B**) Whole cell lysates from melanoblasts 1 h post-irradiation were immunoblotted for the indicated proteins. α-tubulin was used as a loading control. This is representative of 3 independent experiments. (**C**) Levels of phosphorylated ATM and p53, CHK2 and total p53 relative to untreated ATM wild-type melanoblast QF1433. Graphs represent relative band density, normalised to α-tubulin of three independent experiments. Error bars indicate SD. * = *p* value < 0.05; ** = *p* value < 0.01.

**Figure 3 ijms-25-01664-f003:**
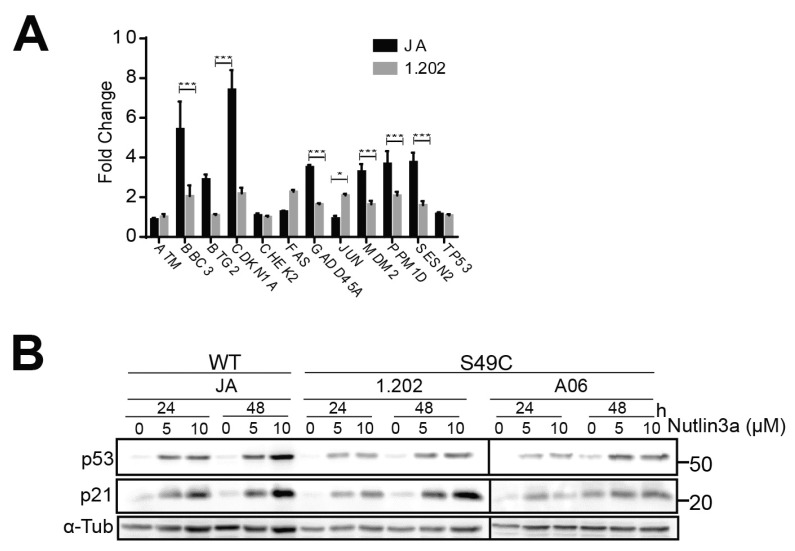
DNA damage increased p53-dependent transcript levels in ATM Ser49Cys variant-expressing cells. (**A**) ATM Ser49Cys demonstrates p53-dependent transcription. Gene expression levels of p53 signalling pathway in JA (WT) and 1.202 (Ser49Cys) 2 h post-irradiation. Fold change is treated over untreated, normalised with house-keeping genes BACT and GAPDH. (**B**) LCL lines were treated with 0, 5 or 10 μM Nutlin3a for 24 or 48 h. Whole cell lysates were immunoblotted for p53, p21 and α-tubulin as loading control. * = *p* value < 0.05; *** = *p* value < 0.001.

**Figure 4 ijms-25-01664-f004:**
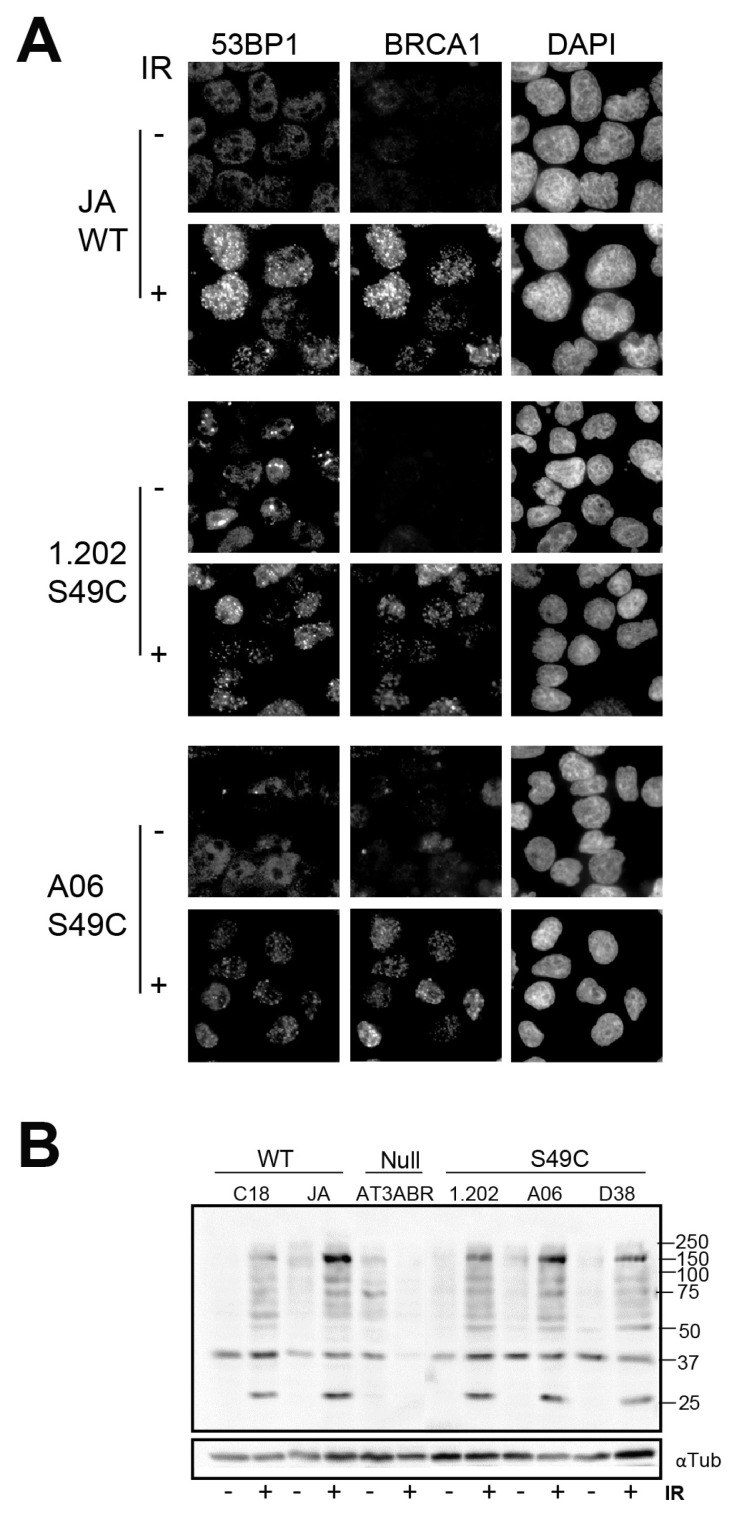
(**A**) ATM Ser49Cys does not affect repair foci formation. Immunofluorescence staining of LCL fixed 2 h with or without irradiation (6 Gy) and probed for 53BP1 and BRCA1. DAPI stained the DNA. (**B**) LCL lines were irradiated as in Figure 1. Whole cells lysates prepared 2 h post-irradiation were immunoblotted for p-ATM/ATR substrates and α-tubulin as a loading control.

**Figure 5 ijms-25-01664-f005:**
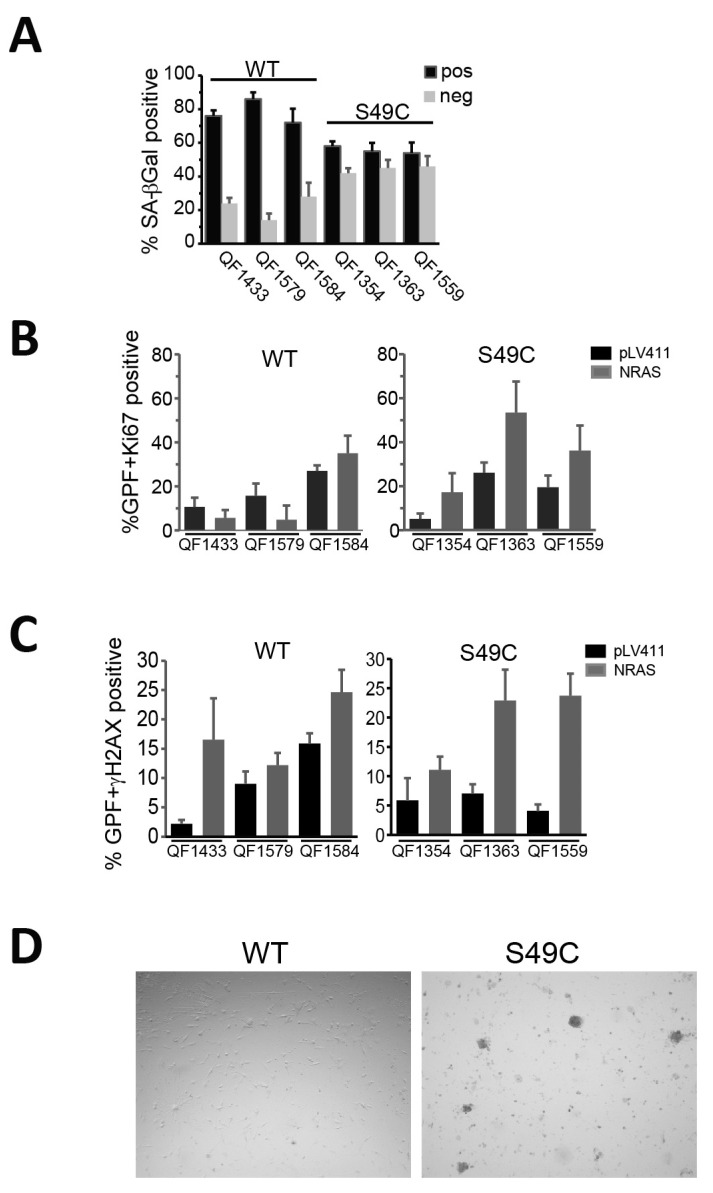
ATM Ser49Cys delays senescence by oncogenic NRAS. (**A**) The percentage of SA-β-Gal-stained cells in the NRAS Q61K-transduced primary melanoblast cell lines 2 weeks after transduction. The data are the mean and SD of >100 cells per well for 6 wells for each cell line. (**B**) High-content image analysis of melanoblasts transduced with NRAS Q61K, vector control (pLV411) or mock for 2 weeks. The percentage of GFP-expressing cells (marker of transduced cells) with high-level Ki-67 staining are shown. This is the average and SD counting up to 3400 cells per well in 6 wells. (**C**) High-content imaging as in (**B**) for γH2AX-stained cells. (**D**) Brightfield images of NRAS Q61K-transduced melanoblasts 4 weeks after transduction.

**Table 1 ijms-25-01664-t001:** Analysis of GeneATLAS (http://geneatlas.roslin.ed.ac.uk/ (accessed 25 November 2021)) for the traits defined below for rs1800054 SNP. * *p* < 0.05.

Trait	Eff. Allele	Beta	pv	MAF	HWE	ORbeta *
C43 Malignant melanoma of skin	C	0.0003	0.6618	0.0147	0.1305	1.05
D03 Melanoma in situ	C	−0.0004	0.3289	0.0147	0.1305	0.83
C43-C44 Melanoma and other malignant neoplasms of skin	C	0.0019	0.2599	0.0147	0.1305	1.05
Skin colour	C	0.0007	0.8603	0.0147	0.1305	-

## Data Availability

Data is available within the paper.

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
