# Peer review of "The ATM Ser49Cys Variant Effects ATM Function as a Regulator of Oncogene-Induced Senescence"

_ijms, 2024, doi:10.3390/ijms25031664_

Round 1
Reviewer 1 Report
Comments and Suggestions for Authors
The authors has revealed that ATM allele has little discernible effect on the acute response to DNA damage. This is an important finding, however several changes may be needed to improve the presentation and quality of the manuscipt:
1. Line 91. Please describe why B95-8 EBV was specificly used in this study.
2. Line 93 - How do you define unaffected relatives? what parametes did you use?
3. Line 117 - what was the does used for irradiation?
4. Line 102 - Please describe the immunofluoresence or immunoblot procedure, then cite the reference paper.
5. Figure 1 A, B, D, Figue 2 A,B - would be better to also show the molecular ladder lane.
6. Line 176 - "Multiple downstream ATM targets", please mention.
7. Figure 4A - would be better to provide the colour image.
Author Response
We thank the reviewer for their comments.
- Line 91. Please describe why B95-8 EBV was specifically used in this study.
This was strained used with success previously to immortalise peripheral blood lymphocytes in the cited reference (ref 12).
- Line 93 - How do you define unaffected relatives? what parameters did you use?
The unaffected relatives where those with no disease as defined in the cited reference from co-authors Aideen M. McInerney-Leo and Emma L. Duncan (ref 10).
- Line 117 - what was the does used for irradiation?
6 Gy γ-irradiation was used as stated in line 100.
- Line 102 - Please describe the immunofluoresence or immunoblot procedure, then cite the reference paper.
We have added more detailed description of the procedures as requested.
- Figure 1 A, B, D, Figue 2 A,B - would be better to also show the molecular ladder lane.
We have added molecular weight markers that correspond to the uncut immunoblots shown in the supplementary data for the same figures. Because the immunoblots are cropped we could only add one that was close the protein band, and thus could not add a useful marker for ATM which was above the highest molecular weight marker of 250kDa.
- Line 176 - "Multiple downstream ATM targets", please mention.
We have assessed in some depth two ATM targets, CHK2 and p53. We have also assessed 53BP1 focus formation, an ATM-dependent process in response to ionising radiation induced DNA damage, and used an antibody direct against ATM phosphorylated substrates in Figure 4. These are characterised in the text referring to this data.
- Figure 4A - would be better to provide the colour image.
We find with green and red images that the grey scale images provide the highest contrast for visualisation. The images are all collected as grey scale images and only false coloured by the imaging software.
Reviewer 2 Report
Comments and Suggestions for Authors
The reviewed article presents research on the potential impact of the ATM Ser49Cys variant on the processes of carcinogenesis and cell senescence. The authors used a very interesting experimental panel that does not raise serious concerns. In addition to molecular studies, the paper also includes an analysis of the influence of the ATM Ser49Cys variant on the risk of melanoma development (analysis of data from the Australian Melanoma Genome Project and Brisbane Naevus Morphology Study). In my opinion, the research direction is intriguing, although before publication, please consider the following comments:
1. Please justify the choice of cell lines (lymphoblastoid and melanoblast cells) in the paper. Why did the authors not decide to conduct studies on commercially available melanoma cell lines?
2. Please provide a detailed description of the statistical methods used in section 2. Additionally, on bar charts, please include markers indicating statistical significance, if present between the compared samples.
3. The paragraph "Cell culture" contains information regarding cell exposure to UVB radiation. Please verify this matter and specify which analyses were performed after UVB irradiation.
4. In the methodology description, the authors often refer to previous publications. From the readers' perspective, it would be beneficial to introduce brief descriptions of the execution of individual experiments.
5. Please provide the dilutions at which antibodies were used.
6. Considering the subject matter of the paper and the multifaceted significance of the ongoing cytophysiological processes, it would be worthwhile to perform an analysis of apoptosis and the cell cycle (at least in selected samples).
7. In the introduction, I suggest describing the background and significance of the senescence process.
Author Response
We thank the reviewer for their comments.
- Please justify the choice of cell lines (lymphoblastoid and melanoblast cells) in the paper. Why did the authors not decide to conduct studies on commercially available melanoma cell lines?
We have used lymphoblastoid and melanoblast cell lines of defined ATM genetics as these are not tumour lines and therefore do not have the many other genetic changes that the tumour cell lines contain. As these are different for each cell lines, it is more difficult to determine whether the changes in response to irradiation is due to the variant being studied. We had attempted to make isogenic ATM Ser49Cys lines in HT1080 to have a direct comparison in an identical genetic background as this would be most accurate comparison, but we were unsuccessful despite being able to knock-in a marker alteration. In the absence of this isogenic model, using non-cancer lines without multiple somatic genetic differences was the best option. This justification is presented of the results section 3.1.
- Please provide a detailed description of the statistical methods used in section 2. Additionally, on bar charts, please include markers indicating statistical significance, if present between the compared samples.
We have added a description as requested in the Methods section. We have not presented extensive statistics of the differences between control and treated the multiple quantitations except for Figure 1C, as although we did see significant increases with treatment in most cases, the main thrust of the work is that there were sufficient differences between different cell lines with the same ATM genotype to make the difference between individual cell lines with different genotypes meaningless. Having significance shown in all cases would obscure the real result which was that despite the difference observed between cell lines, overall there was little difference between ATM genotypes. For this reason we have not provided the requested statistical analysis as we feel it distracts the reader from this result.
- The paragraph "Cell culture" contains information regarding cell exposure to UVB radiation. Please verify this matter and specify which analyses were performed after UVB irradiation.
We have removed this reference to UVB exposure as it is not relevant to this manuscript.
- In the methodology description, the authors often refer to previous publications. From the readers' perspective, it would be beneficial to introduce brief descriptions of the execution of individual experiments.
We have added further experimental detail as requested.
- Please provide the dilutions at which antibodies were used.
Antibody dilutions are dependent on multiple factors and as such this detail can be misleading to readers. We have provided a range of dilutions used for each technique, immunoblotting and immunofluorescence as this provides a truer guide to the reader than specific dilutions for each antibody and technique.
- Considering the subject matter of the paper and the multifaceted significance of the ongoing cytophysiological processes, it would be worthwhile to perform an analysis of apoptosis and the cell cycle (at least in selected samples).
Gross cell cycle changes and apoptosis in response to ionising radiation are only observed at longer time points (>16 h) than used in the present study. We sought to assess immediate downstream effects and direct targets of ATM, hence the analysis of direct phosphorylation targets p53, CHK2 and 53BP1, and analysis using a generic ATM phosphorylation substrate antibody. As we could observe no significant difference in these immediate ATM targets, we have not assessed more downstream outcomes of these target phosphorylations e.g. cell cycle and apoptosis changes.
- In the introduction, I suggest describing the background and significance of the senescence process.
We have added a short section on senescence to the Introduction as requested.
In addition to immediate DNA damage responses, ATM as also been demonstrated to have a role in regulating oncogene-induced senescence barrier to transformation [3]. Oncogene-induced senescence is a significant barrier to transformation in melanocytes, particularly as NRAS and BRAF oncogenic mutations are common driver of nevus formation, and also common oncogenic drivers of melanoma [4]. Thus defective ATM function could contribute to bypassing the senescence barrier in oncogene expressing nevocytes.
Round 2
Reviewer 2 Report
Comments and Suggestions for Authors
The authors have revised the manuscript, incorporating changes based on the reviewer's suggestions. Nevertheless, in my opinion, the omission of the results of the statistical analysis (or selective presentation) does not ensure the credibility of the work. Statistical significance depends not only on the difference between the compared values (height of bars) but also on the distribution of results, standard deviation, and sample size. Not all of these elements may be noted on the graph.
Furthermore, the methodology of the study should ensure full reproducibility of the conducted research and analyses. For this reason, the authors should provide the dilutions at which antibodies were used.
Author Response
We added in the requested statistical analysis to the bar graphs in Figures 2 and 3, and added in the dilution for the antibodies in the methods.
Round 3
Reviewer 2 Report
Comments and Suggestions for Authors
The Authors improved the manuscript according to the suggestions. I recommend publishing the article in its present form.